# Direct Single-Cell Analysis of Human Polar Bodies and Cleavage-Stage Embryos Reveals No Evidence of the Telomere Theory of Reproductive Ageing in Relation to Aneuploidy Generation

**DOI:** 10.3390/cells8020163

**Published:** 2019-02-16

**Authors:** Kara Turner, Colleen Lynch, Hannah Rouse, Vimal Vasu, Darren K. Griffin

**Affiliations:** 1School of Biosciences, University of Kent, Giles Lane, Canterbury CT2 7NJ, UK; k.j.turner-24@kent.ac.uk (K.T.); d.k.griffin@kent.ac.uk (V.V.); 2Cooper Genomics Nottingham, Medicity, D6 Building, Thane Road, Nottingham NG90 6BH, UK; colleen.lynch@coopersurgical.com (C.L.); rouseh@live.com (H.R.); 3Department of Child Health, East Kent Hospitals University Foundation NHS Trust, William Harvey Hospital, Ashford TN24 0LZ, UK

**Keywords:** telomere length, reproductive ageing, first polar bodies, blastomeres, aneuploidy

## Abstract

Reproductive ageing in women, particularly after the age of 35, is associated with an exponential increase in the proportion of chromosomally abnormal oocytes produced. Several hypotheses have attempted to explain this observation, including the ‘limited oocyte pool’ hypothesis and the ‘two-hit’ hypothesis, the latter explaining that a depletion in oocyte quality with age results from the multiple opportune stages for errors to occur in meiosis. Recently however, the telomere theory of reproductive ageing in women has been proposed. This suggests that shortened telomeres in oocytes of women of advanced maternal age render oocytes unable to support fertilization and embryogenesis. Despite a credible rationale for the telomere theory of reproductive ageing in women, very few studies have assessed telomere length directly in human oocytes or preimplantation embryos. Therefore, we directly assessed relative telomere length in first polar bodies and blastomeres from cleavage stage (day 3) embryos. In both cell types we tested the hypothesis that (1) older women have shorter telomeres and (2) chromosomally abnormal (aneuploid) gametes/embryos have shorter telomeres. In all cases, we found no evidence of altered telomere length associated with age-related aneuploidy.

## 1. Introduction

The concept of reproductive ageing in women is well recognized, and it manifests as a sharp decline in natural conception beyond 35 years of age, coupled with an increase in miscarriage rate and an increase in chromosome abnormalities in the conceptus [1]. However, causal mechanisms remain the subject of continued investigation [2]. Reports that an age-related decline in the uterine environment led to a failure to recognize and abort trisomic conceptuses [3] were largely discredited. Rather, it became clear that the underlying cause was a decline in the ability of the oocyte to produce a viable pregnancy [2]. In support of this, oocytes from women of advanced maternal age, defined as over 35 years, are known to be prone to meiotic errors leading to gains or losses of whole chromosomes (aneuploidy). Furthermore, advancing maternal age is a significant contributing factor towards the presence of aneuploidies in preimplantation embryos, with 90% of imbalances attributed to maternal origin in couples where the woman is of advanced maternal age [1,4,5]. Aneuploidy is the leading cause of pregnancy loss in conceptuses and is associated with stillbirth, obstetric complications, imprinting syndromes, infertility and in vitro fertilization (IVF) failure [1,4,5].

Different hypotheses have attempted to explain the observed decline in oocyte potential with advancing maternal age. The ‘limited oocyte pool’ hypothesis proposes that female fertility declines alongside a depletion in the number of oocytes, as well as a decline in the quality of oocytes [2,6,7,8,9]. The ‘two hit’ hypothesis states that a depletion in oocyte quality with age results from the multiple opportune stages for errors to occur in meiosis. That is, errors may occur prenatally during pre-meiotic mitotic divisions, during early stages of meiosis I in prolonged diplotene (which lasts from fetal life until the onset of puberty at least) or during ovulation in the latter stages of meiosis I and/or early meiosis II. Therefore, bivalents that are susceptible to errors might arise prenatally during meiosis I (and are therefore age-independent) followed by age-dependent abnormal processing of the bivalent [10]. More recently however, the ‘telomere theory’ of reproductive ageing has been proposed [11], theorizing that shortened telomeres in oocytes of women of advanced maternal age render oocytes unable to support fertilization and embryogenesis [11]. This telomere shortening may be the result of a combination of the end replication problem in mitotically active precursor cells during gestation [12,13], and prolonged exposure to oxidative stress during the interval between oogenesis and ovulation [14]. Given the important role of telomeres during synapsis and recombination/segregation events during meiosis (which are in turn crucial in ensuring faithful segregation of chromosomes), it is plausible that shortened telomeres in the oocyte may lead to aneuploidy in the mature ovum, and in turn, aneuploidy in the resulting embryo following fertilization. Thus, it is possible that telomeres may play a role in reproductive ageing in women [11,15,16,17,18,19]. The theory is also supported by mouse model data which show that telomerase deficient mice appear to mirror the characteristics of this phenomenon [18,20,21]. Furthermore, women that have given birth to infants affected by trisomy 21 (Down syndrome) possess shortened telomeres compared to age matched controls [22].

Despite a credible rationale for the telomere theory of reproductive ageing, very few studies have assessed telomere length directly in human oocytes or preimplantation embryos and to date, there is no evidence to support the notion that oocyte or cleavage-stage embryo telomeres shorten as women age, or that the telomeres of gametes that undergo segregation errors are shorter than those that do not. This paucity of data is largely due to restrictions in access to sample material, and the technical challenges of measuring telomeres in single cells. Furthermore, available data report conflicting results. While Keefe et al. showed that oocyte telomere length is predictive of embryo development potential [17], Turner et al. found that oocyte telomere length was not associated with IVF outcome or maternal age [23]. In addition, to the best of our knowledge, only one study has assessed telomere length in female gametes and cleavage-stage embryos in relation to the presence of aneuploidy. This study showed that telomere length is significantly reduced in aneuploid polar bodies and cleavage stage embryos compared to sibling euploid polar bodies and cleavage stage embryos respectively from the same IVF cycle [24]. However, details addressing telomere length in polar bodies and embryos in relation to maternal age were absent from this paper. 

Therefore, the aim of this study was to utilize a quantitative real-time PCR (qRT-PCR) methodology (applicable to single cells) to measure the relative telomere length in first polar bodies (a by-product of meiosis I that is representative of genetic material within the oocyte) and single blastomeres from cleavage stage embryos. We employed the standard comparative method for fold-change in relative telomere length between a reference and an unknown sample (shown in Equation (1)) [25]. The purpose was to study the effects of telomere length on reproductive ageing in women, and to specifically test the following hypotheses:
Telomere length is significantly shorter in (a) the first polar bodies and (b) blastomeres from women of advanced maternal age (over the age of 35) compared to their younger counterparts.Telomere length is significantly shorter in (a) first polar bodies, and (b) blastomeres that are involved in chromosome segregation errors compared to sibling euploid first polar bodies and embryos.


## 2. Materials and Methods

The study was conducted with institutional research ethics approval (University of Kent Sciences Faculty Research Ethics Advisory Group) and informed maternal consent to blastomere and polar body biopsy procedures, analysis of chromosome copy number and the use of surplus sample material for research purposes. 

For the assessment of telomere length in first polar bodies and embryos, whole genome-amplified (WGA) DNA from biopsied blastomeres and first polar bodies that were surplus to requirement were donated from couples undergoing IVF treatment (at Care Fertility, across three clinics worldwide) and aneuploidy screening (performed at Genesis Genetics Nottingham, UK) between 2011 and 2013. Leftover WGA material was then transported to the University of Kent (UKC) for PCR purification, and relative telomere length analysis by qRT-PCR. 

A total of 82 polar bodies were assessed from 25 patients, and a total of 86 embryo biopsies were assessed from 22 couples. Blastomeres from cleavage stage embryos were biopsied at the eight cell stage on day three post-fertilization. Relative telomere length in euploid blastomeres and polar bodies were compared to relative telomere length in aneuploid blastomeres and polar bodies, respectively. In addition, relative telomere length in polar bodies and blastomeres from women aged 35 years or younger were compared to those of woman of advanced maternal age (defined as over the age of 35) [26].

### 2.1. Whole Genome Amplification and Aneuploidy Screening

Polar bodies and blastomere biopsies underwent WGA using the SurePlex DNA Amplification System (Illumina), and they were assessed for the presence of aneuploidy using the SurePlex 24 Sure kit (Illumina, San Diego, CA, USA) by CL and HR at Genesis Genetics Nottingham, UK. Procedures for each of these methods were followed according to manufacturer’s instruction. Briefly, biopsies were collected in 2.5 µL PBS in a sterile 0.2 mL PCR tube and 5 µL extraction buffer was added (prepared as per manufacturer instructions). Tubes were then incubated at 75 °C for 10 min followed by 95 °C for 4 min. Following this, 5 µL preamplification cocktail (prepared according to manufacturer instruction) was added to each tube, and samples were incubated at 95 °C for 120 s, followed by 12 cycles of 95 °C for 15 s, 15 °C for 50 s, 25 °C for 40 s, 35 °C for 30 s, 65 °C for 40 s and 75 °C for 40 s. Finally, 60 µL amplification mix was prepared as per manufacturer instruction and added to each sample. Tubes were then incubated at 95 °C for 120 s, followed by 14 cycles of 95 °C for 15 s, 65 °C for 60 s and 75 °C for 60 s. DNA amplification was assessed via gel electrophoresis, and only samples generating an amplification product were labelled using the Fluorescent Labelling system and co-hybridized with labelled control male DNA on to 24 Sure bacterial artificial chromosome microarrays. The resulting 24 Sure microarrays were hybridized, washed, and scanned according to the manufacturer’s instructions. Images were analyzed, quantified, and whole chromosomal copy number ratios were reported using the Cytochip algorithm-fixed settings in BlueFuse Software (BlueGnome, Ltd, Cambridge, UK). Determination of specific SurePlex amplification on the resulting array CGH plot in BlueFuse was by visualization of Y nullisomy in the resulting profiles. Once specific amplification was observed, autosomal profiles were analyzed for the gain or loss of whole chromosomal ratios. Sample profiles were then reported as being either euploid or aneuploid.

### 2.2. Whole Genome Amplification Product Purification

WGA products from polar bodies and blastomeres were purified using the QIAquick PCR Purification kit (Qiagen, Hilden, Germany) according to the manufacturer’s instructions. Briefly, buffer PB was added at five times the product volume to each product, followed by 10 µL 3 M sodium acetate (Sigma-Aldrich, Gillingham, UK). Each sample was loaded onto a QIAquick spin column and centrifuged at 13,000 rpm for 60 s. The flow-through was discarded, and 750 µL of buffer PE was added to each column before two further rounds of centrifugation at 13,000 rpm for 60 s, with the flow through discarded between rounds. Finally, each column was assembled into a clean DNA-free Eppendorf tube and the purified product was eluted in 30 µL buffer EB. DNA was quantified using a Nanodrop spectrophotometer, and the purity was assessed by evaluating the A260/280 and A260/230 values. Those that fell within acceptable ranges (between 1.8 and 2, and between 2 and 2.2 respectively) were stored at −20 °C until use in the qRT-PCR assay for telomere length analysis in single cells.

### 2.3. Quantitative Real-Time Polymerase Chain Reaction Analysis of Relative Telomere Length

In order to assess average relative telomere length, a qRT-PCR protocol originally described by Cawthon 2009 was adapted and optimized for use with the Rotor-gene Q real-time PCR machine (Qiagen, Hilden, Germany). In this reaction, relative telomere length is measured by determining the factor by which the DNA sample of interest differs from the reference DNA sample in its ratio of telomere sequence copy number (T) to the copy number of the reference gene (S) [27]. A multicopy reference gene described by Treff et al. was selected for use, instead of a single copy reference gene (which is described by Cawthon) in order to reduce the effects of allele drop-out and to compensate for differences in chromosome copy number [24]. The design of each primer is shown in Table 1. 

qRT-PCR was carried out in 25 µL reaction volumes, each containing: 2× Rotor Gene-Q SYBR Green PCR mastermix (Qiagen, Hilden, Germany), 100 nM of each primer (either TelG and TelC or AluF and AluR, Eurofins MWG Operon, Ebersberg, Germany) and 25 ng DNA. The sample DNA was diluted to 5 ng/µL in buffer EB immediately prior to qRT-PCR. A total of 5 µL of buffer EB was added in the place of DNA to the no-template negative control tubes. The telomere sequence was amplified under the following cycling conditions: An initial hold temperature at 95 °C for 5 min, two cycles respectively at 94 °C and 49 °C for 15 s each to allow for annealing and extension of TelG, 40 cycles of 94 °C for 15 s, 62 °C for 10 s, and 74 °C for 10 s to enable TelC annealing and extension of TelG products. In a separate reaction, the multicopy reference gene was amplified under the following cycling conditions: An initial hold temperature at 95 °C for 5 min, 32 cycles of 95 °C for 15 s, 60 °C for 15 s and 77 °C for 10 s. A melt curve was performed following each reaction, to ensure reaction specificity, and a two-fold serial dilution of a reference DNA sample (AMS Biotechnology Ltd, Abingdon, UK) was assayed in triplicate in each run, in order to ensure a reaction efficiency of between 95–105%, with an R^2^ value above 0.95. The reproducibility and repeatability of the assay was confirmed by assessing the percentage covariance within and between experiments [28]. The intra- and inter-assay variability were 0.6% and 4.6%, respectively.

Each DNA sample was assessed in triplicate. The amount of telomere sequence present was assessed by averaging the triplicate cycle threshold values for both the telomere and the reference gene reactions, before using the standard comparative method (2^−ΔΔCt^) to determine the fold-change in telomere copy number (representative of the relative telomere length) between the reference sample and the unknown sample [25]. This was calculated by using the formula in Equation 1:

Fold change = 2^−((unknown tel Ct − unknown alu Ct) − (ref tel Ct − ref alu Ct))^(1)


Equation (1): Comparative method for fold-change calculation between a reference and an unknown sample. ‘tel’ refers to the telomere sequence amplification reaction, ‘alu’ refers to the multicopy sequence reaction, ‘ref’ refers to the reference DNA amplification and ‘Ct’ is the cycle threshold.

### 2.4. Statistical Analyses

Statistical analyses were performed using IBM SPSS Version 25. Data were tested for normality using a Shapiro–Wilk test, and results revealed a non-normal distribution (*p* < 0.05). Therefore, non-parametric tests were used for data analyses as follows: A Mann–Whitney U test was employed for unpaired analyses, i.e., comparison of relative telomere length in polar bodies and blastomeres in women aged 35 or under, versus relative telomere length in polar bodies and blastomeres in women aged over 35. A paired samples sign test was employed for paired analyses, i.e., comparison of euploid versus aneuploid polar bodies, and blastomeres originating from the same donors.

## 3. Results

Anonymized patient details and the results of aneuploidy screening (carried out by staff at Care Fertility Nottingham, UK and Cooper Surgical Nottingham, UK) can be found in Appendix A for the first polar bodies and blastomeres respectively. Overall, the relative telomere length was found to be highly variable in the polar bodies and blastomeres assessed. 

### 3.1. Relative Telomere Length in the First Polar Bodies of Younger (≤35) Versus Older (>35) Donors

In order to test the hypothesis that telomere length is significantly shorter in the first polar bodies from women of advanced maternal age compared to their younger counterparts, relative telomere length was successfully assessed in 2–4 first polar bodies from the above cohort of women (Appendix A). An average relative telomere length for all polar bodies assessed in each woman was calculated, and then the data was divided into two groups; those aged 35 or under, and those aged over 35 and therefore defined as being of advanced maternal age. The polar bodies assessed were made up of a mixture of both euploid and aneuploid samples. The data was analyzed for statistical significance using a Mann–Whitney test, and the results are summarized in Figure 1.

Results indicate that, overall, the relative telomere length in the first polar bodies of women of advanced maternal age was shorter than that of women aged 35 years old or under (0.11 compared to 0.2 respectively, Figure 1); however, this difference was not statistically significant (*p* = 0.35). A scatter plot of maternal age versus relative telomere length (shown in Figure 2) showed no correlation between the two (Pearson’s correlation coefficient r = 0.07, *p* = 0.52).

### 3.2. Relative Telomere Length in Cleavage Stage Embryos From Younger (≤35) Versus Older (>35) Mothers

To test the hypothesis that telomere length is significantly shorter in embryos derived from women of advanced maternal age compared to their younger counterparts the above was repeated, but in single biopsied blastomeres. That is, relative telomere length was assessed in single blastomeres biopsied from a total of 44 embryos from 11 couples in which the woman was 35 years old or under and a total of 42 blastomeres were assessed for telomere length from 11 couples in which the woman was of advanced maternal age (Appendix A). From each woman in each cohort an equal number of aneuploid and euploid blastomeres were assessed for relative telomere length (i.e., two aneuploid and two euploid). An average relative telomere length for all blastomeres assessed from each couple was calculated and then the data was divided into two groups; those from women aged 35 or under and those aged over 35 and therefore defined as advanced maternal age. Blastomeres assessed were made up of a mixture of both euploid and aneuploid samples. The data was analyzed for statistical significance using a Mann-Whitney test and the results are summarized in Figure 3.

Results show that relative telomere length in blastomeres biopsied from cleavage stage embryos from couples in which women were 35 years old or under are shorter than those from couples in which women were over the age of 35 (0.13 compared to 0.2 respectively, Figure 3). However, this difference was not statistically significant (*p* = 1.0).

To confirm this finding, relative telomere length measured from each blastomere was plotted against maternal age (Figure 4). Results showed that, in line with results in Figure 3, no correlation existed between the two (r = 0.16, *p* = 0.14).

Overall, evidence shows that although telomere length appeared to be slightly increased in blastomeres derived from cleavage stage embryos from couples in which the woman is of advanced maternal age, this difference was not statistically significant. Therefore, telomere length in the cleavage stage embryo does not appear to be related to maternal age.

### 3.3. Relative Telomere Length in Aneuploid Versus Euploid Polar Bodies

In order to study the relationship between telomere length and aneuploidy in first polar bodies, we tested the hypothesis that telomere length is significantly shorter in first polar bodies that are involved in chromosome segregation errors compared to euploid first polar bodies. Controlling for maternal age and natural variation in inter-individual telomere lengths, a paired analysis was performed by comparing the average telomere length of euploid polar bodies from each woman, to the average telomere length of aneuploid polar bodies from the same women. Average telomere lengths for aneuploid or euploid first polar bodies were calculated from the individual telomere lengths of two aneuploid and two euploid first polar bodies, respectively. Therefore, only those patients from whom two aneuploid and two euploid first polar bodies were available were included in this analysis (*n* = 12 women, total of 24 euploid and 24 aneuploid first polar bodies). The results showed that the average telomere length of aneuploid first polar bodies (0.11) was identical to average telomere length in euploid first polar bodies (0.11) (Figure 5), and a paired samples sign test revealed no statistically significant difference (*p* = 0.39). 

When the data were sub-divided according to maternal age, relative telomere lengths remained identical in euploid polar bodies from women aged 35 years or younger, in comparison to women of advanced maternal age (0.11, *p* = 0.89) as shown in Figure 6. In aneuploid polar bodies, relative telomere length was longer in women aged 35 years or younger, compared to women of advanced maternal age (0.17 compared to 0.10, respectively) as shown in Figure 7; however this difference was not statistically significant (*p* = 0.16).

### 3.4. Relative Telomere Length in Aneuploid Versus Euploid Blastomeres

To test the hypothesis that telomere length is significantly shorter in aneuploid blastomeres compared to euploid blastomeres from sibling embryos, telomere length was assessed in blastomeres derived from aneuploid embryos and compared to that of euploid embryos. Controlling for natural inter-individual variation in telomere length as well as maternal and paternal age, we compared relative telomere lengths in sibling embryos generated in the same IVF cycle. From each couple, two euploid and two aneuploid embryo samples were available; therefore the average telomere length of the two euploid embryos was compared to the average telomere length of the two aneuploid embryos. A total of 84 embryos from 21 couples were assessed. A paired samples sign test was utilized in order to test for a statistically significant difference in relative telomere length in aneuploid embryos, compared to sibling euploid embryos. Results indicated that relative telomere length in aneuploid embryos compared to sibling euploid embryos is variable between cases. While some aneuploid embryos possess shorter telomeres than sibling euploid embryos, others possess longer telomeres. Overall, the telomere length of aneuploid embryos (0.14) compared to euploid embryos (0.18) was not statistically significantly different (*p* = 1.0). This is illustrated in Figure 8.

Since we were unable to account for the effects of other variables that might influence telomere length in these embryos (including paternal age effects), we did not further sub-divide and analyze these data according to maternal age.

## 4. Discussion

In this study, we assessed relative telomere length in the first polar bodies and blastomeres of cleavage stage embryos using a qRT-PCR methodology that was specifically designed for use with WGA DNA from single cells. We investigated the relationship between maternal age and relative telomere length in these cells, and whether shortened telomeres are involved in chromosome segregation errors during meiosis and early cleavage events. Our data showed that telomere length was highly variable in first polar bodies and blastomeres of cleavage stage embryos, in keeping with other published data [23,29,30]. This is also in keeping with other available data, which shows that telomere length is known to be highly variable among all other study populations assessed to date, including fetus’, newborns, infants and adults [31,32,33,34,35,36,37]. Our data additionally indicate that advanced maternal age (>35 years) is not associated with a difference in polar body or embryo telomere length, in comparison to younger maternal age (≤35 years). Furthermore, our results also demonstrate no correlation between telomere length and increasing maternal age in polar bodies and embryos. Finally, no differences in relative telomere length were observed between aneuploid and euploid first polar bodies and blastomeres. 

Our observation that advanced maternal age is not associated with a difference in the relative telomere length of first polar bodies is in line with previous observations using quantitative fluorescence in situ hybridization (QFISH) analysis in oocytes. Therefore, our analysis of relative telomere length in 82 polar bodies adds further weight to previous findings in a study that assessed 26 oocytes [23]. It is possible that this finding can be explained by the maintenance of telomere length in the oocyte by telomerase activity. Indeed, available data indicate that telomerase is expressed in the oocyte at all stages of oogenesis in humans [38] and other mammals [39,40]. Furthermore, it is also possible that telomere length could be maintained in the oocyte by homologous recombination-based telomere lengthening mechanisms, known as alternative lengthening of telomeres (ALT), though evidence to support this is currently lacking in the published literature. 

Alternatively, it is important to recognize that our results may be the consequence of the narrow age range of the mothers sampled in this study (26–47 years; age range 21 years). The majority of samples in our study were from women in their 30s, with only a few individuals outside of this age range. Other studies that have shown a relationship between telomere length and chronological age in a variety of cell types from both men and women have generally included a broader range of ages, from infancy and childhood to old age [37,41,42]. Therefore, the age range available for inclusion in our own study may not have been powerful enough to show such an affect. Moreover, all women included in this study were undergoing fertility treatment, and therefore may not be representative of a normal population. For ethical reasons it is not possible to obtain female gametes from healthy, fertile women and therefore the limitation that we describe in our methodology applies to all published studies in this area. It is possible that a proportion of women included were pursuing fertility treatment for male factor infertility, and therefore may be considered as being healthy and fertile; however, data regarding the cause of fertility was not collected.

To the best of our knowledge, our finding that advanced maternal age does not affect telomere length in cleavage stage embryos is novel, and it appears to be specific to the cleavage stage embryo. Data from a previous study by Mania et al. showed that telomere length is reduced in embryos derived from women of advanced maternal age at day five post-fertilization. However, only 35 embryos from seven couples were assessed in Mania’s study, whereas in the present study, a total of 84 embryos from 22 couples were assessed. Furthermore not all embryos developed at the same rate in Mania’s study, and thus blastocysts, morulae, and arrested embryos were included in the analysis. Therefore the contribution of developmental delay on telomere length cannot be excluded [30], which has previously been related to reduced telomere length in day 3 embryos [17]. 

Although our finding that advanced maternal age does not affect telomere length in cleavage stage embryos is in keeping with our initial observation that maternal age does not affect telomere length in first polar bodies, the results may additionally be affected by the paternal contribution to telomere length in the developing embryo. Several studies have previously identified that paternal age is positively correlated with sperm telomere length [23,41,43], and that telomere length in the offspring is paternally inherited [44,45,46]. However, details of paternal age in the present study were not available, and therefore they could not be assessed in relation to embryo telomere lengths. Alternatively, mechanisms of telomere lengthening in the developing embryo may explain the results obtained here. Although it is generally accepted that telomerase activity becomes most active at the blastocyst stage of development, leading to increased telomere length [22,24,26,28,29,38], it is possible that recombination based mechanisms of telomere lengthening may play a role in earlier cleavage stage embryos [47]. While this has not been demonstrated in human embryos to date, it has previously been observed in mice [47]. 

Our finding that relative telomere length is not altered between aneuploid and euploid first polar bodies is in contrast to previous observations by Treff et al., who showed that telomere length is reduced in aneuploid compared to sibling euploid polar bodies. However, in Treff’s analyses, only nine polar bodies were assessed [24], whereas in our own study, we assessed 82 polar bodies from 25 couples. One possible explanation for the observation found here could be that, while shortened telomeres may result in impaired synapsis and recombination events leading to aneuploidy, such an event may be limited to those chromosomes with critically short telomeres [48]. Since qRT-PCR analysis cannot determine the telomere lengths of individual chromosomes, it is possible that overall relative telomere length is not reflective of critical telomere length that is sufficient to cause aneuploidy in a selection of chromosomes [48]. Furthermore, it is possible that the high level of natural variation observed in telomere length in our study could have masked any effects that could be related to ploidy status. Telomere length is also known to be heterogeneous among different chromosomes, and therefore once again, it may be that natural variability in telomere lengths of the specific chromosome involved in aneuploidy generation contributed to the overall observations in the present study [49]. Alternatively, should telomerase or ALT be active in the oocyte, as suggested earlier, telomere lengths in polar bodies and oocytes may be stabilized. If true, the events leading to aneuploidy must occur by some means other than those directly resulting from shortened telomeres, e.g., inefficient checkpoint mechanisms during meiosis [50,51].

Finally, our results also show that telomere shortening is not related to chromosome segregation errors in the cleavage stage embryo. This finding is in contrast to that which has previously been shown by Treff et al., whom showed that telomere length is reduced in aneuploid cleavage stage embryos in comparison to sibling euploid embryos. However in that study, only 18 embryos (nine aneuploid and nine euploid) from a total of nine couples were assessed [24], whereas in the current study, 84 embryos (42 aneuploid and 42 euploid) from a total of 21 couples were assessed. Thus, it is possible that the discrepancies in these data might be due to the limited sample size in the previous study. Further and as previously discussed, it is possible that the high variability in telomere length observed here and by others may have masked any effects relating to aneuploidy generation in embryos. Another explanation for the findings of our own study and that of Treff et al. lies in the assumption that a single blastomere from each embryo is representative of the entire embryo. Many studies before this have identified that embryos (and in particular cleavage stage embryos) are prone to mosaicism [52,53,54], in which daughter cells are not genetically identical. Previous studies have also shown that telomere length is not identical among different blastomeres of the same embryo [23]. Thus, it is possible that the blastomeres sampled from the embryos included in this study may not have been reflective of the embryo as a whole. Unfortunately, it was not possible to obtain all blastomeres from any embryo in the present study and therefore, this phenomenon could not be explored. 

The strengths of our study lie in the fact that this is the largest study to date to assess relative telomere length in polar bodies and blastomeres from cleavage stage embryos; a total of 82 polar bodies and 86 blastomeres were assessed for relative telomere length, which was averaged across 25 and 22 couples, respectively. The data presented here adds further information to what is currently published, and additionally offers the benefits of the inclusion of solely polar bodies from good quality oocytes and blastomeres from good quality embryos, as assessed by a trained embryologist. However, it is possible that our data are limited by technical factors in the methodology used. Due to the nature of the samples, (WGA DNA from biopsied first polar bodies and blastomeres from cleavage stage embryos), it was not possible to analyze telomere length by the classic telomere restriction fragment (TRF) analysis, because this requires at least 1–2 µg of intact DNA. The process of WGA produces long fragments of DNA representing the whole genome, but it is impossible for the DNA to remain intact in its native form. It was also impossible to use QFISH as others have done, as this technique requires intact whole cells, whereas our samples were lyzed prior to WGA. Therefore, we chose to use qRT-PCR to assess telomere length in single cells [24]. Traditionally, relative quantitation of telomere length using qRT-PCR takes advantage of a single copy reference gene [27,55]; however, we chose not to use a single copy reference gene methodology due to the well-characterized issues surrounding single locus bias following WGA [56]. In addition, the use of a multicopy reference gene is thought to reduce bias resulting from the presence of aneuploidy, which could result in the calculation of artificially short TS ratios in cases where missing chromosomes would lead to fewer telomere repeat copies, or artificially long TS ratios in cases where extra chromosomes would lead to extra telomere repeat copies [24]. However, the possibility of under- or over-representation of one or both target sequences cannot be excluded. Previous observations suggest that there are inter-individual differences in the copy numbers of Alu repeats [57,58], and therefore, this inter-individual heterogeneity in Alu copy number could conceivably impact upon subsequent relative telomere length quantification. In future studies, it may be prudent to identify and to assess a different multicopy reference loci that is not susceptible to inter-individual variability. Alternatively, the inclusion of more than one reference locus may help to reduce this effect.

Finally, it would have been instructive to have established the aetiological cause of infertility in our study cohort in order to include this information in our analyses and conclusions. However, access to patient clinical notes was not a requirement for aneuploidy screening, and therefore these were not provided at the time of sample submission. It was not possible retrospectively to request patient consent for these notes, due to the time frame in which the samples were collected (between 2011 and 2013). Having said that, in many cases, the reasons for infertility are multifactorial, and in some cases (15–30%), the reason for infertility is undefined (idiopathic) [59]. As a result, we considered that even if it was possible to request patient consent for access to clinical notes, it would be unlikely that this would reveal any meaningful trends in the sample population included in our study.

## 5. Conclusions

To the best of our knowledge, this is the largest study to directly address telomere length in human female gametes, and the first to do the same in human cleavage stage embryos in relation to maternal age. It is also the largest study to address telomere length in female gametes and cleavage stage embryos in relation to chromosome segregation errors. Previous studies have shown that reduced telomere length is associated with chromosome instability [60], impaired synapsis and recombination in female gametes in the mouse model [18,61], which is in turn linked to aneuploidy [18] in the embryo. However, the results presented here suggest that telomere shortening is not related to female reproductive ageing in humans. 

## Figures and Tables

**Figure 1 cells-08-00163-f001:**
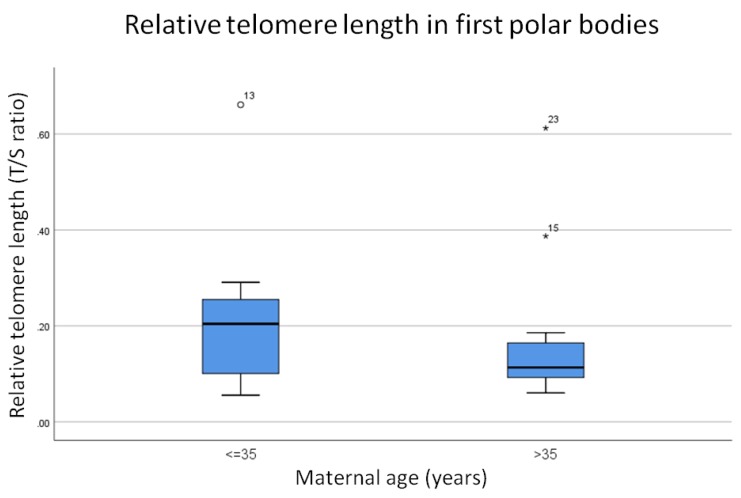
Average relative telomere length of all first polar bodies from women aged 35 years old or under (*n* = 13) compared to women of advanced maternal age (*n* = 12). Results show that telomere length is slightly shorter in first polar bodies from women above 35 years old, however, this difference is not statistically significant (*p* = 0.35). The line on the lower end of the whisker represents the minimum T/S ratio, the bottom of the box represents the first quartile, the line through the middle region of the box represents the median T/S ratio, the top of the box represents the third quartile, and the line at the top of the whisker represents the maximum T/S ratio. Circles represent outliers between 1.5 and 3 box lengths from the upper region of the box. Asterisks represent extreme outliers that are more than 3 times the interquartile range from the third quartile.

**Figure 2 cells-08-00163-f002:**
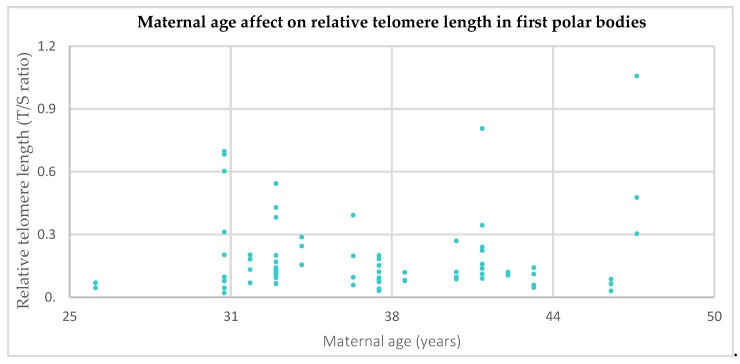
Relative telomere lengths of aneuploid and euploid first polar bodies in relation to maternal age (*n* = 82 first polar bodies from *n* = 25 women). Results indicate no correlation between relative telomere length of first polar bodies and maternal age (*p* = 0.52).

**Figure 3 cells-08-00163-f003:**
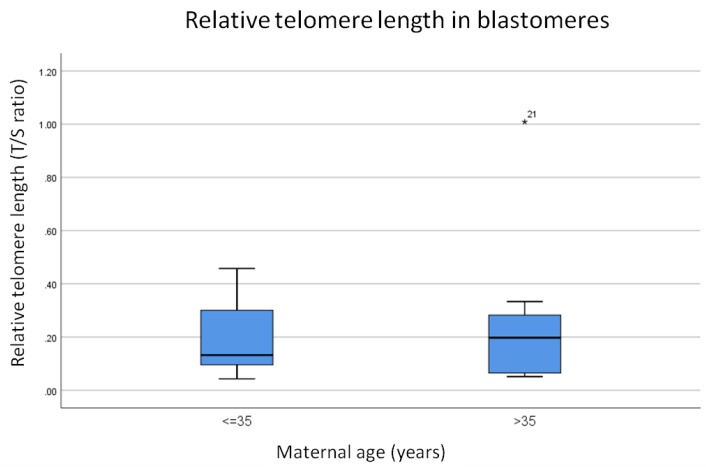
Average relative telomere length in all blastomeres from younger women (*n* = 11) compared to those from women of advanced maternal age (*n* = 11). Relative telomere length is shorter in blastomeres derived from couples in which women were younger however this difference was not statistically significant (*p* = 1.0). The line on the lower end of the whisker represents the minimum T/S ratio, the bottom of the box represents the first quartile, the line through the middle region of the box represents the median T/S ratio, the top of the box represents the third quartile and the line at the top of the whisker represents the maximum T/S ratio. Circles represent outliers between 1.5 and 3 box lengths from the upper region of the box. Asterisks represent extreme outliers that are more than 3 times the interquartile range from the 3rd quartile.

**Figure 4 cells-08-00163-f004:**
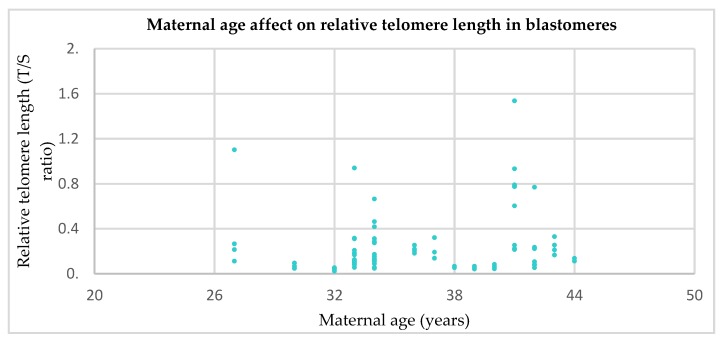
Telomere length in blastomeres from cleavage stage embryos in relation to maternal age (*n* = 86 blastomeres from *n* = 22 couples). Results show no correlation (*p* = 0.14).

**Figure 5 cells-08-00163-f005:**
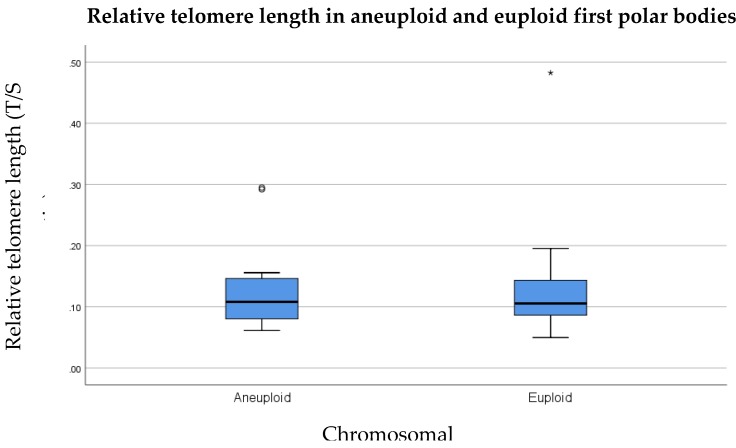
Relative telomere length of aneuploid first polar bodies (*n* = 12) in comparison to euploid first polar bodies (*n* = 12) assessed in all women. Results show no significant difference (*p* = 0.39). The line on the lower end of the whisker represents the minimum T/S ratio, the bottom of the box represents the first quartile, the line through the middle region of the box represents the median T/S ratio, the top of the box represents the third quartile, and the line at the top of the whisker represents the maximum T/S ratio. Circles represent outliers between 1.5 and 3 box lengths from the upper region of the box. Asterisks represent extreme outliers that are more than three times the interquartile range from the third quartile.

**Figure 6 cells-08-00163-f006:**
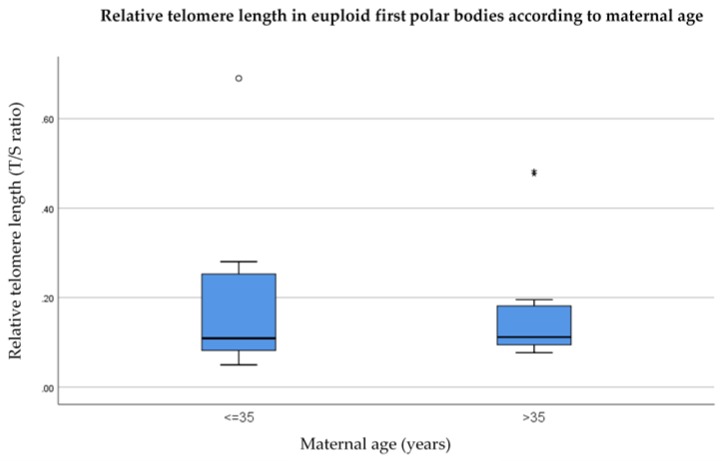
Relative telomere length of euploid first polar bodies in women aged 35 years or younger (*n* = 11) in comparison to woman over the age of 35 (*n* = 11). Results show no significant difference between the groups (*p* = 0.16). The line on the lower end of the whisker represents the minimum T/S ratio, the bottom of the box represents the first quartile, the line through the middle region of the box represents the median T/S ratio, the top of the box represents the third quartile, and the line at the top of the whisker represents the maximum T/S ratio. Circles represent outliers between 1.5 and 3 box lengths from the upper region of the box. Asterisks represent extreme outliers that are more than three times the interquartile range from the 3rd quartile.

**Figure 7 cells-08-00163-f007:**
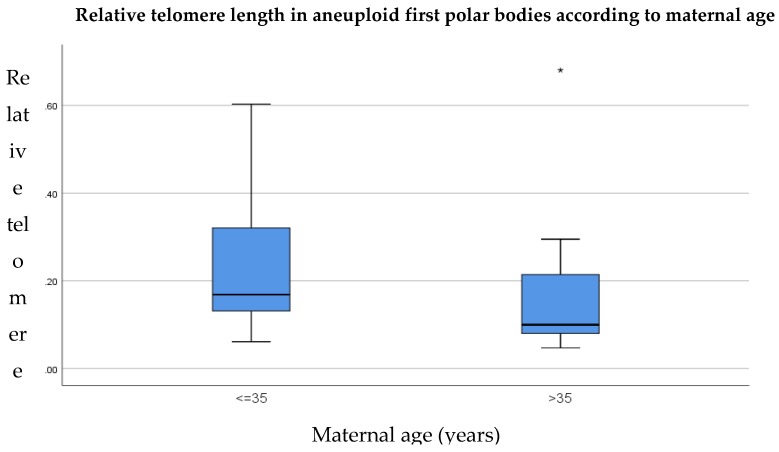
Relative telomere length of aneuploid first polar bodies in women aged 35 years or younger (*n* = 8) in comparison to woman over the age of 35 (*n* = 12). Results show no significant difference between the groups (*p* = 0.89). The line on the lower end of the whisker represents the minimum T/S ratio, the bottom of the box represents the first quartile, the line through the middle region of the box represents the median T/S ratio, the top of the box represents the third quartile and the line at the top of the whisker represents the maximum T/S ratio. Circles represent outliers between 1.5 and 3 box lengths from the upper region of the box. Asterisks represent extreme outliers that are more than three times the interquartile range from the third quartile.

**Figure 8 cells-08-00163-f008:**
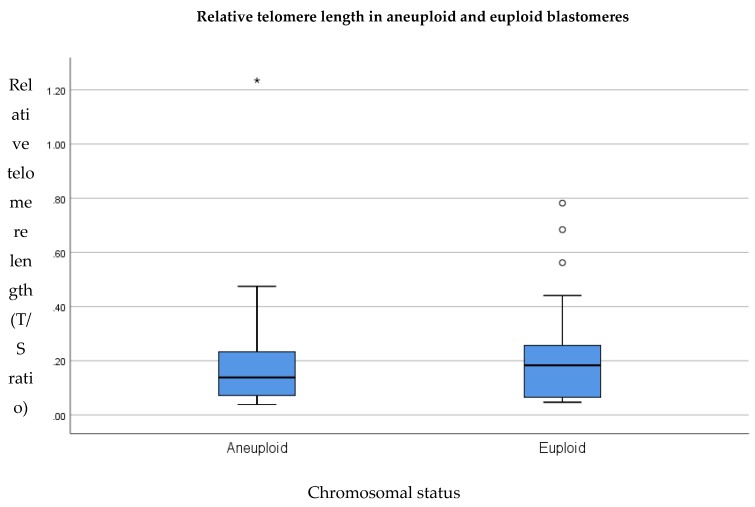
Relative telomere length of euploid (*n* = 21) and aneuploid (*n* = 21) blastomeres assessed from all couples. Results show no significant difference (*p* = 1.0). The line on the lower end of the whisker represents the minimum T/S ratio, the bottom of the box represents the first quartile, the line through the middle region of the box represents the median T/S ratio, the top of the box represents the third quartile and the line at the top of the whisker represents the maximum T/S ratio. Circles represent outliers between 1.5 and 3 box lengths from the upper region of the box. Asterisks represent extreme outliers that are more than three times the interquartile range from the third quartile.

**Table 1 cells-08-00163-t001:** Primer design for amplification of the telomere and the multicopy reference gene for quantitative real-time polymerase chain reaction (qRT-PCR) analysis of the relative telomere length in polar bodies and blastomeres.

Name	Design	Reference
TelG	ACACTAAGGTTTGGGTTTGGGTTTGGGTTTGGGTTAGTGT	[27]
TelC	TGTTAGGTATCCCTATCCCTATCCCTATCCCTATCCCTAACA	
AluF	GACCATCCCGGCTAAAACG	[24]
AluR	CGGGTTCACGCCATTCTC	

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
