# Peer review of "Direct Single-Cell Analysis of Human Polar Bodies and Cleavage-Stage Embryos Reveals No Evidence of the Telomere Theory of Reproductive Ageing in Relation to Aneuploidy Generation"

_cells, 2019, doi:10.3390/cells8020163_

Round 1
Reviewer 1 Report
In the paper by Turner et al. the relative telomere length in human oocytes and single blastomeres donated by women undergoing IVF treatment has been assessed and correlated with the aneuploidy observed in the same samples and with the donor age.
This was done to test the hypothesis that telomere shortening could be related to reproductive ageing and aneuploidy, as suggested by studies in model organisms and still debated in human studies.
The work expands previous studies that showed a reduction of telomere length in aneuploid polar bodies and cleavage stage embryos by enlarging the number of analysed individuals and addressing telomere length in relation to age.
The experimental approaches are reliable, and results are clearly reported. The results do not support the idea that short telomeres are associated with aneuploidy or maternal age; the possible reasons for these negative results are thoroughly discussed in the discussion section.
Globally, the paper adds valuable information on the analysed issue and warrants publication in the journal provided that the following points are addressed:
1. The rationale for measuring telomere length in blastomeres, beside oocytes, should be explained in the introduction
2. Figures: Report in figure legends the number of analysed individuals in the two groups. Explain the elements of the box plot: are the line depicting the median and the box quartiles? What average means? What “upper region” of the box means? Orientate properly the Y label of figures.
3. Introduction: reference 19 does not support the content of the sentence.
4. Pag. 3 chapter 2.2. references are not properly cited. Telg , Telc should be TelC, TelG. The indication of Celsius grade is superscript; The sentence explaining the equation should be reported as part of the introduction. The number of the sentence is 1 or 2.1?
5. At which cleavage stage were the blastomeres sampled? (8 cell stage?)
Author Response
Reviewer 1
Globally, the paper adds valuable information on the analysed issue and warrants publication in the journal provided that the following points are addressed:
The rationale for measuring telomere length in blastomeres, beside oocytes, should be explained in the introduction
Thank you for this comment which we overlooked in our original draft. In this study, we investigated the ‘telomere theory’ of reproductive ageing in women which suggests that telomere shortening in the oocyte (as a result of the end replication problem in mitotically active precursor cells and prolonged exposure to oxidative stress) leads to impaired synapsis and aneuploidy in the mature ovum (explained in lines 57-66). This would also result in the production of an aneuploid zygote upon fertilisation. In light of this, we have now included the following in lines 60-65 “Given the important role of telomeres during synapsis and recombination/segregation events during meiosis (which are in turn crucial in ensuring faithful segregation of chromosomes), it is plausible that shortened telomeres in the oocyte may lead to aneuploidy in the mature ovum and in turn, aneuploidy in the resulting embryo following fertilization. Thus, it is possible that telomeres may play a role in reproductive ageing in women [11-16].”
2. Figures: Report in figure legends the number of analysed individuals in the two groups. Explain the elements of the box plot: are the line depicting the median and the box quartiles? What average means? What “upper region” of the box means? Orientate properly the Y label of figures.
Thank you very much for this comment. We have now updated the figure legends to explain the elements of the box plots and the number of individuals assessed, as requested. We have also orientated the Y axis correctly in those graphs in which the title was not displayed in the correct orientation.
Introduction: reference 19 does not support the content of the sentence.
Many thanks for this comment. We have removed this reference from the introduction and the reference list.
4. Pag. 3 chapter 2.2. references are not properly cited. Telg , Telc should be TelC, TelG. The indication of Celsius grade is superscript; The sentence explaining the equation should be reported as part of the introduction. The number of the sentence is 1 or 2.1?
We would like to thank the reviewer for spotting these errors in the manuscript. We have now amended the text with the correct citation for telomere primer names and we have indicated degrees Celsius as a superscript. We have also amended the reference to equation 1 in line 180 and included an explanation of the method of relative telomere length quantification in the introduction in lines 86-88.
5. At which cleavage stage were the blastomeres sampled? (8 cell stage?)
Thank you very much for highlighting that the stage of development at which embryos were biopsied has not been clarified within the manuscript. We have now amended the methods section in line 108-109 as follows: "Cleavage stage embryos were biopsied at the 8-cell stage on day three post fertilization.”
Reviewer 2 Report
The manuscript by Turner et al. aims to test the hypothesis of telomere contribution to reproductive ageing in relation to aneuploidy generation in humans. The manuscript is well-written, the topic is interesting and the data are valuable, still I have several comments and questions:
1) The donors of the material were patients undergoing IVF treatment. I understand that no other material is available and it is great that the team took the opportunity to study this important question. The authors point to limits of the study performed on the potentially biased sampling. Nevertheless, it would be better if they considered at least the reason for infertility. The authors stated that „data regarding cause of fertility was not collected“. Why? Could they be obtained at this stage?
2) The age is a continuous variable; I think that it is much better to treat it like it rather than to test differences between two arbitrarily defined age categories (women aged 35 years or younger versus woman of advanced maternal age, defined as over the age of 35).
3) I most cases, the authors ignored pseudoreplications. They compared 82 polar bodies and 86 embryo biopsies from 25, respectively 22 couples, and also two euploid and two aneuploid samples per a donor in the paired test. A statistical control for the pseudoreplications (repeated measures design, mixed-models…) should be performed.
4) The detected outliers were excluded from the comparisons. Did it influence the results of statistical analyses?
5) It is not important for polar bodies, but embryos could be of different sex and X and Y sex chromosomes might differ in telomere length. Maybe, sexes can also differ in Alu copies (the Alu loci are used as reference in qRT-PCR for telomere measurements). Would it be possible to sex the embryos and to check if sexual differences bias the analyses?
6) The authors measured telomeres in triplicates, but they do not present any analysis of repeatability of telomere length. Could they perform a test of correlation between telomere measurements in polar bodies and embryos? If this correlation is not strong, paternal contribution to telomere length might be substantial and might obscure the picture in mothers (or the measurements are not very precise).
7) More information how aneuploidy was detected is needed.
8) I have some doubts about the procedure of qRT-PCR measurements of telomere length. The authors argue that due to potential biased caused by Whole Genome Amplification from single cells, a multilocus marker (Alu) is better than a conserved single copy reference gene. However, this procedure might cause several biases: there are inter-individual differences in copy number of Alu repeats and it might be thus not the best choice to use Alu repeats as a standard; Alu copies can be missing - or on the other hand can be added - in aneuploids (it depends whether reference loci are linked or not to the chromosome involved in aneuploidy – could it be tested?). One solution would be to test several single copy reference loci instead and to compare the results.
9) I found strange that the authors did not find any significant differtences between aneuploid and euploid samples. Even if the loci used as a reference in qRT-PCR are linked to chromosomes not involved in aneuploidy, an addition or a loss of telomeres of a chromosome should be measurable. Or is the procedure used for measurement of telomere length so imprecise? Please, comment on it.
Author Response
Reviewer 2
The manuscript by Turner et al. aims to test the hypothesis of telomere contribution to reproductive ageing in relation to aneuploidy generation in humans. The manuscript is well-written, the topic is interesting and the data are valuable, still I have several comments and questions:
1) The donors of the material were patients undergoing IVF treatment. I understand that no other material is available and it is great that the team took the opportunity to study this important question. The authors point to limits of the study performed on the potentially biased sampling. Nevertheless, it would be better if they considered at least the reason for infertility. The authors stated that „data regarding cause of fertility was not collected“. Why? Could they be obtained at this stage?
The authors would like to thank reviewer 2 for this comment. We agree that it may have been useful to consider the reasons for infertility, however this was not possible and we are happy to explain the reason for this as follows: The samples that were selected for inclusion in the study and processed for telomere length analysis, were leftover WGA DNA samples that had been sent to Genesis Genetics, Nottingham for aneuploidy screening (as described in the methods section). The biopsies from which the WGA DNA material was obtained were originally performed at a variety of clinics worldwide (and then sent to Genesis Genetics Nottingham for aneuploidy screening) and therefore there was no clinical information provided with the samples (this is not required for aneuploidy screening). In order to obtain this information, it would have been necessary to request the clinical notes from the various clinics from which the samples were produced, which would involve contacting each couple in order to request consent for clinical information to be disclosed and included in the study (albeit anonymously). The samples included were from clinical cases between 2011 and 2013, and telomere length analysis was performed in 2014. Therefore, at the time that telomere length measurements were collected, it was deemed that many of the couples (particularly those from 2011/2012) would no longer be in attendance at the clinic and therefore, it may prove difficult to reach out to these couples to obtain consent. Subsequently, with new GDPR regulations coming into effect in 2016, we did not anticipate that it would be possible nor appropriate to request further clinical information from these couples, as this would involve de-anonymising the samples, the information for which should not still be in existence. Finally, in many cases the reasons for infertility are multifactorial and in some cases (15-30% depending on the literature source referred), the reason for infertility is undefined (idiopathic) (Gelbaya, T.A., Potdar, N., Jeve, Y.B. and Nardo, L.G., 2014. Definition and epidemiology of unexplained infertility. Obstetrical & gynecological survey, 69(2), pp.109-115.). Since our sample size included 22 and 25 couples for polar body and blastomere biopsies respectively, we considered that even if it was possible to request patient consent for access to clinical notes, it would be unlikely that this would reveal any meaningful trends.
In light of this comment, we have now updated our manuscript in lines 462-471 to include this explanation and cited the above reference ([59] in the manuscript). We hope that this is satisfactoryto the editor.
2) The age is a continuous variable; I think that it is much better to treat it like it rather than to test differences between two arbitrarily defined age categories (women aged 35 years or younger versus woman of advanced maternal age, defined as over the age of 35).
Many thanks for this comment. During the design of the study, we pondered as to how best to analyse the data in relation to maternal age. We chose to analyse by categorical data for the following reasons:
It is very well-defined within the literature that women of advanced maternal age (i.e. over the age of 35 years) experience a significant decline in the ability to conceive naturally coupled with a significant increase in the rate of miscarriage and aneuploidy conceptuses. Therefore, it seems reasonable to test our hypothesis by categorising our subjects into one of two groups (35 years and younger and over 35 years old).
Similar categorical approaches have been carried out by other studies that have investigated the effects of reproductive ageing in women and therefore we sought to make our own study comparable to these in order to add information to the currently available literature.
We would add that in addition to our categorical analysis, we have also provided correlation data which maternal age as a continuous variable which also shows no relationship with increasing maternal age.
We hope that this helps to clarify our reasoning and that both the Reviewer and Editor will accept the justification for our methodology.
3) In most cases, the authors ignored pseudoreplications. They compared 82 polar bodies and 86 embryo biopsies from 25, respectively 22 couples, and also two euploid and two aneuploid samples per a donor in the paired test. A statistical control for the pseudoreplications (repeated measures design, mixed-models…) should be performed.
Thank you very much for this comment regarding the handling of our data, which we are happy to clarify. We agree that it is important to erradicate pseudoreplications in the data set and to ensure independence of samples, both within and between groups (except for when appropriate paired analyses are performed in the latter case). In the manuscript we describe calculation of an average TS ratio for all samples of the same subject that are to be included within the same group. For example, in lines 198-204, we state that “In order to test the hypothesis that telomere length is significantly shorter in first polar bodies from women of advanced maternal age compared to their younger counterparts, relative telomere length was successfully assessed in 2-4 first polar bodies from the above cohort of women (table 1). An average relative telomere length for all polar bodies assessed in each woman was calculated and then the data was divided into two groups; those aged 35 or under and those aged over 35 and therefore defined as advanced maternal age.” We have employed the same approach for all of our data analysis (described in lines 234-238, 273-277 and 321-324), which ensures that each subject is only included once within groups (and therefore ensuring independence of samples where appropriate). We deemed that this was the best approach to employ, since a repeated measures design is useful for repeated measured across groups (e.g. use of the same subjects more than once longitudinally or in response to different test conditions), but not within groups. Furthermore, a repeated measures approach also stipulates that data should be normally distributed and that variance should be equal, both of which assumptions were violated in our data.
While we appreciate that calculating an average of the repeated samples from each subject may artificially weaken any trends present, we found that however we tested our data, the result was always acceptance of the null hypothesis (p=>0.05) and therefore, although p value may be weakened, the overall outcome remains the same.
Having said this, we do recognise that figures 1 and 3 were not representative of how the data had been handled in the original draft. This was an oversight and we have now updated these figures and included the number of subjects within the figure legends at the request of reviewer 1. We have also updated lines 434-437 in the discussion, to make this approach clear to the reader as follows: “The strengths of our study lie in the fact that this is the largest study to date to assess relative telomere length in polar bodies and blastomeres from cleavage stage embryos; a total of 82 polar bodies and 86 blastomeres were assessed for relative telomere length, which was averaged across 25 and 22 couples respectively.” We hope that this helps to clarify the type of analyses that we have performed and to justify our approach.
4) The detected outliers were excluded from the comparisons. Did it influence the results of statistical analyses?
We are very grateful that reviewer 2 has brought this query to our attention. Please accept our sincere apologies. Our statement suggesting that we had excluded outliers from our analyses was incorrect. We had considered excluding outliers in an earlier draft of the manuscript and had since come to the decision that, since these data were not linked with any erroneous sample handling, data collection or analyses, it was important to include them in order to draw meaningful overall conclusions about our data. Unfortunately, we overlooked this statement in our submitted draft of the manuscript and have now removed this from line 196. That said, we did perform analysis both including and excluding outliers. In both cases, although the p values were different, the null hypothesis was accepted (i.e. p = >0.05).
5) It is not important for polar bodies, but embryos could be of different sex and X and Y sex chromosomes might differ in telomere length. Maybe, sexes can also differ in Alu copies (the Alu loci are used as reference in qRT-PCR for telomere measurements). Would it be possible to sex the embryos and to check if sexual differences bias the analyses?
The authors would like to thank reviewer 2 for this question, which is an excellent point and something that we had not considered. However unfortunately, although in theory it may have been possible to obtain this information at the time that the embryos were screened, this information was missing from the aneuploidy results provided by Genesis Genetics at the time and therefore we are unable to identify the sex of the embryos.
6) The authors measured telomeres in triplicates, but they do not present any analysis of repeatability of telomere length. Could they perform a test of correlation between telomere measurements in polar bodies and embryos? If this correlation is not strong, paternal contribution to telomere length might be substantial and might obscure the picture in mothers (or the measurements are not very precise).
Many thanks for this comment, which we had also overlooked in our original version of the manuscript. Of course, assay reproducibility is an important factor to consider and therefore we have now updated lines 172-174 to state that “The reproducibility and repeatability of the assay was confirmed by assessing the percentage covariance within and between experiments (Bustin et al 2009). The intra- and inter-assay variability were 0.6% and 4.6% respectively.” We did not perform a test correlation between telomere length measurements in polar bodies and embryos, because the polar bodies assessed were not related to the embryos produced: That is, they originated from different mothers. We apologies for the confusion here, but we add that a relationship between polar bodies and embryos is not eluded to within the manuscript. Therefore, the type of analysis suggested by reviewer 2 would not have been appropriate. We hope that the revision to the methods section (as described above) is satisfactory to address this comment.
7) More information how aneuploidy was detected is needed.
Thank you for this comment, which we have now addressed in lines 115-135 of the Methods section in the manuscript. This includes further information regarding both whole genome amplification and aneuploidy detection in polar bodies and blastomeres.
8) I have some doubts about the procedure of qRT-PCR measurements of telomere length. The authors argue that due to potential biased caused by Whole Genome Amplification from single cells, a multilocus marker (Alu) is better than a conserved single copy reference gene. However, this procedure might cause several biases: there are inter-individual differences in copy number of Alu repeats and it might be thus not the best choice to use Alu repeats as a standard; Alu copies can be missing - or on the other hand can be added - in aneuploids (it depends whether reference loci are linke”d or not to the chromosome involved in aneuploidy – could it be tested?). One solution would be to test several single copy reference loci instead and to compare the results.
Thank you for your query highlighting the use of a multicopy reference locus instead of a single copy reference gene, as described in the original version of the protocol for telomere length analysis by qRT-PCR (described by Cawthon et al., 2002). Our rationale for the use of a multicopy reference locus was akin to that described by Treff et al. 2011, who previously measured relative telomere length in polar bodies and embryos. That is, we wished to eliminate problems arising as a result of single locus bias during whole genome amplification and any bias that the presence of extra or missing chromosomes may have had on the TS ratio calculated. For example, the presence of extra chromosomes (that may not contain the loci of a single copy reference gene) would result in extra copies of the telomere sequence overall, and in turn, TS ratios that were artificially long in comparison to euploid samples. Similarly the absence of chromosomes would result in fewer copies of the telomere sequence overall, resulting in TS ratios that appeared artificially short in comparison to euploid samples. We wished to avoid this effect and therefore designed our study to include a multicopy reference sequence. In so doing, we anticipated that any change in TS ratio observed would be more likely to be related to the generation of chromosome segregation errors (i.e. the cause), and not the result of the number of telomere sequences present overall (i.e. the effect).
Although we had mentioned in our discussion a rationale for the use of a multicopy reference in the context of the affect that single locus bias may have (lines 448-451), we admit, that we overlooked the inclusion of further details relating to bias in TS ratios generated by extra or missing chromosomes. We have now rectified this in our discussion in lines 451-454 in order to address this.
The suggestion to assess several multicopy reference loci is not something that we had considered, however this might have provide some valuable insight and potentially an opportunity to reduce the effects of inter-individual variability in reference locus copy number on TS ratio. Therefore, we have also now edited the discussion section to include the reviewers comment for consideration in future studies (lines 460-461).
9) I found strange that the authors did not find any significant differences between aneuploid and euploid samples. Even if the loci used as a reference in qRT-PCR are linked to chromosomes not involved in aneuploidy, an addition or a loss of telomeres of a chromosome should be measurable. Or is the procedure used for measurement of telomere length so imprecise? Please, comment on it.
We are happy to address this concern from reviewer 2 as follows: Telomere length is known to be highly variable in all sample types and all populations assessed (adult and infant as well as embryos and foetuses) and this is well cited within the literature. We have mentioned in lines 344-346 of our discussion that “Our data showed that telomere length was highly variable in first polar bodies and blastomeres of cleavage stage embryos, in keeping with other published data” and we have now updated this paragraph to detail the same observations found in other study populations (see lines 346-348). It is also known that telomere lengths are heterogeneous among different chromosomes. Taken together, this means that it is possible that any effects on telomere length relating to aneuploidy status may be masked by natural variation that is observed in telomere length among different individuals and among different chromosomes. We have therefore updated lines 407-412 as follows: “Furthermore, it is possible that the high level of natural variation observed in telomere length in our study could have masked any effects that could be related to ploidy status. Telomere length is also known to be heterogeneous among different chromosomes and therefore once again, it may be that natural variability in telomere lengths of the specific chromosome involved in aneuploidy generation contributed to the overall observations in the present study.” We have also updated lines 423-425 to reflect this possibility.
Round 2
Reviewer 2 Report
The authors made a good job in revising their manuscript. They explained several issues and stress now in the manuscript some limitations of their study design. To me, the present form is much more balanced and correct.